# The Patient’s Voice as a Parent in Mental Health Care: A Qualitative Study

**DOI:** 10.3390/ijerph192013164

**Published:** 2022-10-13

**Authors:** Hanna Stolper, Karin van Doesum, Petra Henselmans, Anne Lynn Bijl, Majone Steketee

**Affiliations:** 1Erasmus School of Social and Behavioural Science, Erasmus University Rotterdam, 3062 PA Rotterdam, The Netherlands; 2Jeugd ggz Dimence Groep, 7416 SB Deventer, The Netherlands; 3Department of Clinical Psychology, Radboud University Nijmegen, 6525 XZ Nijmegen, The Netherlands

**Keywords:** integrated family approach, parental mental disorder, adult and child mental health services, infants and early childhood, transmission of psychopathology, family focused practice, qualitative study

## Abstract

Objective: This study is an evaluation of patients in mental health care who have undertaken treatment with an integrated family approach. The treatment focuses on the mental disorders of patients, their role as parents, the development of their young children, and family relationships. The treatment was conducted by professionals from an adult and a child mental health service in the Netherlands. The aim of the study was to identify the key elements and processes of this approach in order to develop a theoretical model. Background: Parental mental disorders have an impact on parenting and child development. To stop detrimental cascade effects and prevent parents and children from being caught up in the intergenerational transmission of psychopathology, a family approach in mental health care is needed. Methods: A qualitative design was adopted using thematic analysis. Data were collected through 18 interviews with patients. The themes in the interviews were which outcomes the patients experienced and which key elements of the treatment contributed to these outcomes. Results: In general, patients were satisfied with the treatment offered. Improved outcomes were within the domain of the family, the parent-child relationship, individual symptoms, and the functioning of the parent and the child. Patients mentioned six key elements of success in treatment: focus on the whole family, flexible treatment tailored to the situation of the family, components of the whole treatment reinforcing each other, multi-disciplinary consultation among involved professionals, a liaison between adult and child mental health services, and attention to the social and economic environment. Conclusions: According to the majority of patients, treatment with an integrated family approach in mental health care is of value for themselves, their children, and family relationships, especially the parent-child relationship.

## 1. Introduction

Being a parent is a meaningful and demanding task, regardless of whether the parent has or does not have a mental disorder. However, it is generally more challenging for the first group, as symptoms can interfere with daily childrearing, and there is a risk of reactivating or exacerbating the symptoms [1,2,3]. Moreover, children are at risk of developing a mental disorder themselves. The transmission of psychopathology from parent to child ranges from 41–77% for the whole diagnostic spectrum [4]. This risk appears to be greater during pregnancy and early life because these phases are crucial for the development of the brain and for building a secure attachment relationship that positively impacts the development of the young child [1,5].

Despite the fact that 12–45% of the patients who receive treatment in mental health care have children [6] and perform the parental role in daily life, little attention has been paid in treatment to their parenting or their children’s development [7]. The underlying reasons for this may be found at the level of the professionals, the patients, and the organization of mental health care, which in many countries is divided into adult and child services. This gap between adult and child mental health services is closely related to the still dominant medical model in which mental disorders are conceptualized as a disease of the individual that should be treated as such. Professionals trained according to the medical model have insufficient knowledge about parenting and children and lack the skills to discuss and integrate these issues into treatment [8]. Although patients do have concerns about the impact of their symptoms on their children [9], they have difficulty bringing this up because of the fear of stigma and outplacement of their children [10]. All these issues constitute significant barriers to an integrated family approach in mental health care practice. Solantaus et al. [11] suggest, and found some evidence in research, that parenting may act as a “facilitator” of patients’ recovery (p. 239). Supporting the parental role and addressing concerns about the patients’ children in mental health care increases patients’ motivation for treatment and results in improved outcomes. It can also provide an opportunity to prevent children from experiencing the intergenerational cycle of mental disorders. Therefore, a paradigm shift in mental health care from an individual to a family approach is needed [7,12].

With the concerns discussed above in mind, an adult mental health service (AMHS) and a child and adolescent mental health service (CAMHS) in the Netherlands have established a liaison to offer parents of young children mental health care treatment with an integrated family approach in which the focus of treatment is on the family as a whole [13]. This involves the treatment of the mental disorders of the parent, the development of their young child as well as the parent-child relationship and other relationships within the family. In a family approach, the other (ex/co)parent is routinely involved, with any conflicts between the partners addressed and incorporated into the treatment. The aim of this integrated family approach in treatment is to increase the quality and efficiency of the treatment for parents and their young children, to improve their relationship, and to ameliorate the risk of intergenerational transmission of psychopathology or other adverse outcomes.

In our previous paper [13], we presented a theoretical model regarding key elements, processes amongst professionals, and benefits of an integrated family approach based on the experiences of professionals from an adult and child mental health service who have integrated their treatments with a focus on the whole family. Based on group interviews, we concluded that treatment with an integrated family approach in mental health care is of value for families because of the focus on the whole family, implementing a flexible complementary treatment plan, and empowering professionals through multi-disciplinary consultations.

In the current study, we will evaluate this integrated family approach in treatment from the parent’s point of view, and, in addition, compare the parents’ experience with the experiences of professionals from the previous study. The aim of this study is to evaluate the integrated family approach regarding patients’ experiences and improve the theoretical model further. In addition, we hope to contribute to the practice of integrated mental health care and inspire and motivate professionals and managers of mental health services to offer parents and their young children integrated treatment. This study is part of a more extensive study of an integrated family approach in the treatment of patients and their young children in mental health care services.

The research questions were: What do patients identify as improved outcomes for themselves and the whole family as a result of an integrated family approach in treatment? Relatedly, how did the perspectives of patients on these improved outcomes compare to those of professionals? Second, what do patients indicate as the key elements of success in the integrated family approach? Third, what challenges or barriers did patients experience that posed a threat to the success of treatment? Fourth, what modifications could be made, based on patients’ accounts of their experiences, to the theoretical model of an integrated family approach based on the experiences of professionals and the literature? By finding answers to these research questions, we aim to further improve the integrated family approach to mental health practice.

## 2. Materials and Methods

### 2.1. Design

We conducted a multiple case study in which we interviewed 18 patients as parents who have undertaken an integrated family approach in their mental health treatment involving both adult and child mental health services. We adopted a qualitative design using thematic analysis [14] because we were interested in the experiences of patients with an integrated family approach in their treatment and comparing their experiences with the experiences of professionals. The method of thematic analysis is suitable for identifying common and overarching themes through open, axial, and selective coding. Atlas-ti 8 software was utilized for coding.

We conducted individual interviews which have the potential to gather differentiated information and increase the chance of deepening the information gathered during the interview.

### 2.2. Data Sampling

This current study is part of a case study evaluating the use of an integrated family approach in treatment with collaborating professionals who conducted the treatment. In order to be able to build a theoretical model in which the experiences of professionals and patients are incorporated, it was essential to consider both this dataset and the larger dataset. A total of 19 families participated in the study, however, one family dropped out due to out-of-home placement. Data consisted of semi-structured interviews with 18 patients who had been treated according to an integrated family approach. Ethics approval was granted by the Medical Ethics Review Board at the University Medical Centre of Utrecht in the Netherlands (18–186/C). Patients who were in treatment with an integrated family approach were asked by their therapist to participate in the study. As for the children, the consent of both parents with custody was necessary and both were informed of the scope of the study and how privacy was guaranteed. The 18 families that were included consisted of families whose treatment has been completed when the study started and who had given their informed consent to participate.

### 2.3. Sample

The selection of the 18 family cases was based on the following criteria, which are consistent with those used in the linked study [13], “adult patients with a mental disorder according to the DSM-5 and a young child up to six years, with relational problems or other disorders according to the DSM-5” (p. 6). “We tried to follow the daily practice of complex cases treated by mental health services very closely. In clinical practice, there is a wide variety in the phenomenology of mental disorders [15] and the contexts of the patients and families. Therefore, this study did not focus on a specific classification to avoid the false impression of a homogeneous group that could have been treated in a uniform way” (p. 9). In all of these families, there was complexity, with problems in different domains (individual and family) being highly interrelated and mutually influencing one another. The parents’ and children’s characteristics are shown in Table 1.

This table shows that the families who had been treated are heterogenous regarding their ages, level of education, family structure, DSM-5 classifications, and duration of treatment they received. Comorbidity is present in 66.7% of the parents and 38.9% of the children. In this sample, 14 parents (77.8%) had been previously treated in mental health care, of whom 12 (66.7%) had received treatment more than twice. An additional notable characteristic is that of the 18 parents, 7 were not raised by their own parents, and 6 of the 18 children had grown up with only one of their biological parents.

In a review of intervention targets in the treatment of parents with mental health disorders and their young children for the purpose of preventing the intergenerational transmission of psychopathology, the social and economic context is mentioned as one of the important domains that should be addressed [16]. In our sample, 67% (N = 12) of the families experienced additional problems in this domain which included housing (N = 5), unemployment of patient (N = 11), absence from work due to mental disorder (N = 4), financial dependency of social services or government (N = 9), and debts (N = 2). In 72.2% (N = 13) of the families, there were two or more agencies involved, in addition to the two mental health services, and in 11.1% (N = 2), there was one other agency. In three of the families, child protection services were involved. In five families, professional support was offered with financial management. In our sample, we had no information about the quality of patients’ neighborhoods and did not collect data about their social networks.

### 2.4. The Treatment

In an integrated family approach, there is no fixed treatment plan. Depending on the problems, existing proven effective interventions are used. In all treatments of the 18 families, at least two of the different treatments mentioned in Table 2 were combined and tailored to the needs of an individual family. Several of these treatments require some explanation. With the term psychotherapy, we include a wide range of psychotherapeutic methods: Mentalization Based Treatment [17], Schema Focused Therapy [18], Guideline Informed Treatment for Personality Disorders [19], Cognitive Behavioral Therapy [20], and Short-term Psychodynamic Supportive Psychotherapy [21]. Systems Training for Emotional Predictability and Problem Solving (STEPPS) is a group training to provide patients with borderline personality disorder with information about the disorder and help them in learning skills to regulate their emotions [22]. In the parent-child psychotherapy [23], and the parent-child group, if possible with both parents and child, the focus is on strengthening the parent-child relationship. Parent counseling with both (ex/co) parents focuses on parenthood and the relationships within the family. In this study, we use the term family therapy if any member of the extended family or other important relationships was involved in the treatment. Home treatment is offered at home with a broad scope, ranging from parenting skills, family relationships, and practical as well as emotional support [24]. Storytelling is a method to treat preverbal trauma in young children. Parents tell their young child the story of the traumatic event from the perspective of the child, while the therapist is tapping the child [25]. 

Table 2 shows that in this sample, the majority of interventions were made in the domain of the individual patient and the parent-child relationship. One-third of the parents were involved in couples’ therapy, and over one-fifth were involved in family therapy.

### 2.5. Procedure

Parents had the option of choosing whether to have the interview at their home (N = 7) or the office of the service (N = 11). The period between completing their treatment and giving the interview was several weeks on average. Prior to conducting the interview, the purpose of the interview and the role of the researchers were explained and anonymity was guaranteed.

The interviews were conducted by the two primary researchers (H.S. and P.H.), who both had completed a multi-day training for qualitative research. They were well informed about the process of an integrated family approach in treatment because they participated as professionals in the multi-disciplinary consultation among the professionals. They did not interview participants with whom they personally had been involved as a therapist. Although the interviewers were familiar with the integrated family approach, they did not read the parents’ and children’s case files to avoid any possible prejudice. The only information they received in preparation was an outline of the whole treatment period which contained the family structure, some data from the anamnesis of both parent and child, the sequences of treatments offered, and other agencies involved during treatment. This outline was created by one of the other researchers who studied the electronic case file of the whole treatment at AMHS and CAHMS. The time span of the interviews ranged from 45 to 90 min. Interviews were semi-structured: interview questions were open, exploratory in their nature, and guided by the research questions and the topic list developed by the researchers. The key questions covered the overall experiences of an integrated family approach, perceived outcomes, and enabling and obstructive elements. Afterward, the participants received a small gift for their child.

The method of constant comparison between the different interviews of patients was used during the process of data sampling. Interviews were audio recorded and transcribed and the text was proofread. After finishing each interview, the interviewer shared their impressions with one of the researchers in a debriefing about the atmosphere, content, observations, and differences with previous interviews. A report was made on the content of these debriefings.

### 2.6. Data Analysis

The transcripts of the interviews were anonymized; not only was the patient identified with a code rather than their name, but also any professionals mentioned by the patient were identified with a code indicating their discipline. This was to prevent the researchers from being influenced by recognizing a specific colleague during the analysis process. The whole process of data analysis was done by the three researchers (H.S., P.H., and A.B.), including coding, memoing, and writing a debriefing after finishing coding for each interview. The memos contained information about shared emerged ideas and perceptions, identified relationships within the data, and were used in searching for patterns. The first step in analyzing the data was for two researchers, including the one who conducted the interview, to read the text of the interviews. Next, we performed open coding of the transcripts in which codes were assigned to quotes that could be meaningful in light of the research questions. This was followed by axial coding, in which common and overarching themes were identified.

To increase inter-coder reliability, the first 13 interviews were coded separately by two individual researchers (H.S./P.H. 5, H.S./A.B. 3, P.H./A.B. 3, H.S. and P.H. with a research assistant). In a joint consultation, they compared their choices with each other and discussed them to achieve consensus. After every coding session, a short report was written about the process of coding, for instance, particular impressions, dilemmas with coding, and the reliability of coding, which was estimated by the degree of consensus. The last five interviews were coded only by the interviewers themselves where cases of doubt were discussed with the other interviewer.

Since the present study was linked to the study on the experiences of professionals, we followed the same criteria of data sufficiency in both studies; after coding the first five interviews, there was a codebook with broadly formulated categories. Data sufficiency [26] was achieved when no significant new information emerged during the interviews and no new codes emerged during the analysis. Given the wide diversity of the cases, we added a few more interviews, in the end, to control for possible bias. The text of the interviews and all codes were read, reread, and compared. From this data, several themes emerged about the patients’ view of an integrated family approach in treatment conducted by adult and child mental health services. These themes include which benefits for the family they perceived, which key elements of this approach they perceived as contributing, and which elements were challenging for them.

## 3. Findings

In this section, we bring the experiences of the adults in mental health care in their role of patient, parent, and partner into focus. The analysis of the data demonstrates that, according to the majority of patients in this sample, using an integrated family approach in treatment generates improved outcomes for them, although there were also negative experiences. We will first present which improved outcomes parents mentioned and compare them with those reported by professionals. Subsequently, we will discuss what parents mentioned as negative aspects of integrated treatment. Second, we will present the six key elements that emerged from the data which are specific for the success of an integrated family approach in treatment, and two nonspecific elements. Third, we will present the six experienced threats, and recommendations mentioned by patients. Fourth, we will compare the identified key elements with the theoretical model presented in the previous study based on the experiences of professionals [13] and merge them into a more comprehensive model.

### 3.1. The Improved Outcomes and Some Negative Experiences of the Integrated Family Approach in Treatment Perceived by Patients

Table 3 shows what parents reported about the improved outcome they experienced in different domains: individual patient (adult and child), the parent-child relationship, and family relations, as well as in how many interviews one or more improvements in the particular domain were reported.

Parents mentioned improvements in all domains, the most in the family domain, followed by the reported improvements in the domains of the individual adult patient and the parent-child relationship.

About two-thirds of the 18 patients were positive about their experiences with an integrated family approach in their treatment. In 15 interviews, the parent said they would recommend this treatment to others in the same circumstances. Three patients reported some mixed feelings: overall positive but negative about some aspects of the treatments. Three patients were less satisfied with the achieved outcome of the treatment in terms of their own individual functioning (confusion about diagnosis and no focus on treatment), the functioning of the partner relationship (unfulfilled desire for the continuation of couples therapy), as well as the progress made by the child (unresolved behavioral problems, the child placed out of home). All three of those patients were diagnosed with an autism spectrum disorder.

We have compared the data of improved outcomes perceived by patients as is presented in Table 3 with the data from a previously conducted study [13] in which we interviewed professionals about what they perceived to be the results in these cases of their treatment with an integrated family approach. Both studies are part of the entire case study in which the same treatments were evaluated with both professionals and patients, respectively, with group interviews and individual interviews. Table 4 shows the comparison of the treatment outcomes in different domains mentioned by parents with those reported by professionals.

From this table, it can be inferred that compared to professionals, parents report more overall improvement as a result of the treatment. Furthermore, it shows that parents and professionals agreed that major improvements occurred for the individual adult patient and improvements in the quality of the parent-child relationship. The main difference in the evaluation of the impact of treatment between parents and professionals is seen in the domain of the child, in which 10 parents reported that the functioning of the child had improved, versus in 3 cases according to professionals. There is also a substantial difference reported in the family domain, in which parents perceived more improvements than professionals did.

### 3.2. Key Elements of an Integrated Family Approach That Contributed to Outcomes

In this section, we will present the key elements patients identified contributing to the improvements they described. An overview of the key elements which contributed to the success of the treatment is presented in Table 5, and these elements are discussed below.

#### 3.2.1. Focus on the Family as a Whole

In total, 14 of the 18 interviewed parents told us that the focus on the family as a whole was helpful, and they were happy about the fact that the treatment was not only focused on their mental disorder or about their young child but that professionals’ attention was also on family relationships and safety within the whole family.

A father: “*What I really appreciate is that there is a more comprehensive approach, so that they don’t just look at it like: ‘Hey, a child with a problem has been brought in and we’re going to do something about it,’ but they consider the whole picture.*”

A mother: “*There was attention focused on me, my development and the development of my child.*”

Some parents recognized that at the start they did not realize there was a reciprocal relation between their own problems or functioning and the functioning of their child. They had not asked for help for the whole family, but it was their therapist who made this connection for them.

A mother with a personality disorder: “*We presumed that we, the father and I, have the problem, not our child. And we never really thought about the fact that such a young child may already have something with trauma. If we hadn’t been told that, we would never have been here with our child.*”

Some parents knew by themselves that it was not only their individual mental disorder, but their problems interacting within their relationships and other problems in the family which therefore required extensive treatment: 


*“The non-functioning was in the whole context, was in our whole family and it all interacts. … That all needed attention and it would have been very skewed if it would have remained only with the treatment of me.”*


Another parent said: “*I went for help because I realized that I had trouble bonding with my daughter after her birth. I thought she didn’t want me. And I never wanted that to become a disturbing factor between us. So, I needed to do something about it.*”

The attention to safety in the family was explicitly mentioned by a father and considered by him to be an important aspect of the whole treatment: 


*“In the beginning, it sometimes escalated to the point where there was some physical violence, that I hit him [child], and then certainly attention was paid to that, because it was the mother who contacted the therapist (CAMHS) about this, and she did respond to that appropriately… ‘now the safety of the kids is the most important thing’.”*


#### 3.2.2. Flexibility of Treatment, Tailored to the Situation of the Family

In 12 of the 18 interviews, parents mentioned the flexibility of treatment, by which they meant that the different treatments of AMHS and CAHMS were timed and tailored to their situation, needs, and wishes, as an important factor. Parents valued that the family’s situation, the family members’ capacity, and the processes in treatment were all considered.

A teenage mother: “*They were very flexible. They were actually very good at helping me where it was needed.*”

In one of the families, there was a 5-year-old son with behavioral problems and problems in the parent-child relationship which were hypothesized to be rooted in the period when the mother had severe postpartum depression and they were separated. For the mother, it was still impossible to see baby photos of her son and be able to listen to the music box melody. Despite the therapists’ recommendation of treatment for the mother, she did not have enough energy for it, but the parents were motivated to have a few couples therapy sessions in which they could work on improving their communication by discussing the pattern that had developed between them during the mother’s depression. She said: 


*“Well the part of treatment [for me], we actually put that aside and the father was allowed to come along. And then we talked more about the fact that he said he still finds it difficult, the feeling that he should stay at home, because I do not feel well…. And that I could openly say [to him] that I am managing, although not having an easy time. … Then I thought, yes, they [therapists] could easily switch to that [couples therapy].”*


A father said: “*It gave me the feeling that people were open and not just thinking, ‘Hey, this is your problem, this is how we treat it, and you have to go through the ‘car wash’ in this way, but that there was also space for individual variation.’*”

Some parents mentioned that the timing of the different treatments, in particular, the timing of attention to parenting and the parent-child relationship, was essential for them. One mother indicated that she could not profit much from the parent-child group because her mind was still so full of sadness from missing her own parents. After this contact was re-established, there was space for a parent-child treatment. 

Another mother said: “*What I really appreciated is that at some point I indicated: ‘I first want to do something about myself and there was just simply space to do so’.*”

Parents mentioned that, in practical terms, treatments were coordinated “*Sometimes I had up to three to four times a week sessions of therapy, but all those appointments were coordinated.*”

#### 3.2.3. Components of the Whole Treatment Reinforced Each Other

In 10 of the 18 interviews, it was mentioned that the treatment at AMHS and CAMHS complemented and reinforced each other. 

A mother: “*I did really need both. Through the one, the other got better and through the other, the one in turn got better.*” A father: “*And I think she [the therapist] did a very good job of making the link between those things that were going on with me, but also between the mother and me, affect that original situation with our child. … Actually, we have been in treatment and now, our child is doing better.*”

A mother with a personality disorder and severe depression with suicidality talked about how the three treatments after a period of hospitalization—individual, couples, and parent-child treatment—reinforced each other. 


*“I had a lot of guilty feelings towards my daughter. I had let her down in a terrible manner and I couldn’t really cope with that. But then by exploring that with my therapist, I could look back on it more easily. That the guilt diminished and that I could also return to the positive work with her, that this was possible. … The conversations with my therapist also reinforced me during the conversations in the couples therapy. Even though I have been sick for a period of time, that does not mean that I am of less value to my children and cannot do this [upbringing] less well. … So that did help me in the conversations in the couples therapy [to be more assertive].”*


A mother and her young daughter were traumatized by the death of, respectively, her partner and stepfather. The daughter received treatment for her trauma in which the mother and therapist (CAMHS) prepared the trauma narrative together (Lovett, 1998). Subsequently, the mother was reading this trauma story to her daughter, while the therapist was by tapping the child. The mother told us the positive effect she experienced for her daughter, herself, and her relationship due to this treatment: 


*“Because it has brought her calm, it has also brought me calm and we have become closer and understand each other better.”*


#### 3.2.4. Multi-Disciplinary Consultations

In 10 interviews, parents mentioned the importance of collaboration and communication in multi-disciplinary consultations between professionals, and transparency about the content with the parents. The benefit of the collaboration and communication between the involved professionals was that they did not have to tell their whole story over and over again, and there was an understanding of the family context: 


*“My therapist knew that I was dealing with quite a special child and that it was tough. And also the reverse, the therapist for my son knew that I had gone through so many traumas and that it could interfere with the current situation.”*


Furthermore, it prevented the family from becoming overburdened: 


*“The communication between all parties was really helpful … there was a good fine tuning … so we didn’t get overloaded.”*


In some cases, in which additional agencies other than mental health services were involved, there was an urgency to establish clarity regarding the contribution and goals of all professionals involved. In one family, the mother, in her desire to improve the family situation, accepted all the help she could find, which led to a situation where she participated in the parent-child group at the mental health service and at the same time was found to be participating in a preventive program where parents and their young child met twice a week in a community center. The mother told us:


*“At one point we had a conversation with the social community service because [they said] ‘we [all involved professionals] just need to be in one line. We need to know from each other who is doing what’.”*


She indicated that afterward, she experienced more relaxation doing things one step at a time.

One father told us that he had more insight into the connections between the different problems in the family due to the feedback he received from the professionals about what was discussed in the multi-disciplinary consultations:


*“Well, the most helpful thing has been the way you have organized it [the integrated family approach] with the different disciplines having a multi-disciplinary consultation… In the beginning, I was not able to see the links [between the problems], and that consultation was very helpful in terms of how it was discussed. I think we were just an agenda item. … The feedback that followed was very helpful for me to say ‘What the hell, but that has a link with what I do myself [and what was discussed there] in relation to my son’.”*


Transparency about the communication between the involved professionals contributes to patients’ confidence in the professionals and the treatment they provide. 


*“And if something had to be done it wasn’t behind my back, everything was just discussed. Only after I had given my consent, that I agreed with it, only then were other connections made.”*


#### 3.2.5. A Liaison between AMHS and CAHMS

Although both services, AMHS and CAHMS, were independent agencies, each with its own workforce, therapeutic view, and workflow, both were part of an overarching organization in which they decided to implement an integrated family approach between the two services. This gave parents the feeling of being treated by one service, which generated clarity and trust for them. Some of the parents mentioned that it meant there was no waiting time for the different treatments at AMHS and CAMHS, while others mentioned they had to tell their story only once. A mother who first started at AMHS and after a period subsequently was at CAMHS: 


*“That you don’t have to start all over again somewhere, but that you already know what it is, just that familiarity.”*


#### 3.2.6. Attention to the Social and Economic Environment

Patients realized that the mental disorder was not an isolated issue and forms part of a network of interrelated problems that need attention. One patient stated that professionals should explicitly ask about the social and economic situations of patients *“because people may feel ashamed to talk about it.”* This patient explained further that the stress caused by these problems may compromise treatment success: *“… and then you tackle one problem, but the energy runs off in another part.”*

In our sample, nine patients told us that their therapist paid attention to environmental problems such as the socioeconomic situation; in seven families, no attention was needed because there were no environmental problems (N = 5), or all such problems were already addressed (N = 2). One patient did not want any attention to it and one patient suggested that attention was necessary despite there being no actual environmental problem. As mentioned before, in 14 families there were other agencies involved, for instance: social work services (housing, finance, administration, practical and pedagogical support, additional childcare, or volunteers for support at home), child protection, or a clinic for psychiatry, obstetrics, and pediatrics.

Regarding the social context, in almost half of our sample, patients told us that extended family was a topic in the treatment. As a consequence of the mental disorder of the parent, support was needed and provided by the grandparents of the child. In one family they fulfilled the role of foster parents, in three families the baby and his parents temporarily lived with the grandparents, and in one family there was a regular sleepover every week. In some cases, the grandparents actually participated in a few (group) therapy sessions and evaluations. In other cases, the relationship with the grandparents was a topic in treatment; for instance, one patient mentioned mixed feelings about their dependency on the support of the grandparents and guarding their own autonomy in relation to the grandparents.

### 3.3. Nonspecific Elements Which Contributed to the Success of Treatment

Nearly all parents mentioned elements contributing to the outcome of the overall treatment which are nonspecific to an integrated family approach. The two most common themes were the positive therapeutic relationship and the use of videotapes.

#### 3.3.1. The Therapeutic Relationship

There was satisfaction with the positive therapeutic alliance in about 12 interviews; in five interviews, there were mixed positive and negative feelings, and in one there was dissatisfaction with the relationship with almost all of the professionals involved. The analysis of the positive and negative comments revealed the following aspects of the professional which parents indicated were important for them: empathetic (understanding, accepting, and repair of mismatches), flexible (appointments and responding to the patient’s current needs), offered continuity in the relationship, transparent, reliable, and the relationship felled equal and connected.

#### 3.3.2. The Use of Videotapes

Although this study did not aim to evaluate specific intervention techniques, some parents spontaneously mentioned this when asked about satisfaction and dissatisfaction with an integrated family approach in their treatment.

In 12 interviews, parents spontaneously mentioned the use of videotapes of the parent-child interaction as a helpful method in the treatments at CAMHS, while two parents with an autism spectrum disorder said the videotapes were not useful for them, and two parents reported that it caused them stress, but was nevertheless very helpful. Videotapes were used in the treatment in two distinct ways. First, at the start of all treatments, a videotape of the interaction in the parent-child dyad was made, and this was reviewed with the parents. This is a part of the assessment of the quality of the interaction within the dyad and reviewing the tape with parents and listening to their observations gives the therapist an impression of the parental reflective functioning. This helps the therapist determine the most appropriate port of entry for treatment. Second, videotapes are used as part of parent-child therapy, the parent-child group, and sometimes in-home treatment, to review with parents and discuss together. From what parents shared, it is apparent that it was experienced as helpful in different ways. Some parents had the experience that their negative perception of their relationship with their child did not always match what they observe in the videotapes: 


*“So I felt like he [child] didn’t like me because I actually didn’t like him in the beginning either. … And because I then watched the tape and realized that, well, how I think it is and how it actually happens are not necessarily the same.”*


Another parent stated: “*What I really appreciated was that I learned how to pick up on very small signs from my daughter, for example that she leaned towards me. I didn’t see all that. I was so negative in my mind.*”

Some parents said they directly benefited from the therapist’s feedback during the reviewing of the tape: 


*“She said I was doing a lot of things well, but that sometimes I was going a bit too fast for him [her son]. That I should adapt to his pace. That was a good thing to get that insight.”*


For some parents, just the awareness of the good things that had changed in the interaction during treatment was a nice experience: 


*“When he stood beside me, that I just held him for a moment, it just came naturally. Then I thought, okay that’s nice (laughs).”*


Some parents mentioned that they became aware of a pattern in the interaction and that this resulted in modification of their behavior. In the parent-child group, there were a few sessions in which the other parent, or another important person, joined. One part of the program is that a videotape is made in which both parents are with the child, and this is reviewed with both parents. In one family with a one-year-old child, the recording showed how the father, with his enthusiasm for playing with the child, received all of the child’s attention and (unintentionally) excluded the mother. Watching the video together brought an awareness of this pattern to both of them. 

The mother said: “*I didn’t notice it at all until we looked over the video. Then it was clear that I was moving more and more backwards and that is rather a confrontation because you do not really notice it yourself.*” She added that, as a result of this, they now do more together as the three of them.

### 3.4. Challenges, Barriers, and Recommendations

Of the 18 parents we interviewed, 11 mentioned dissatisfaction with some different aspects of the treatment; of which, the highest number of comments and suggestions were made by parents with an autism spectrum disorder.

#### 3.4.1. Organization

Although not exactly specific to an integrated family approach, the two most important points of dissatisfaction were the high turnover of professionals during treatment and the time they had to wait for indicated treatment. Because of the latter, in some cases, the treatment at AMHS and CAMHS could not start at the same time, which was perceived as a major barrier.

Furthermore, some parents mentioned they would have appreciated more information at the start about what an integrated family approach in treatment involves. By this, they meant, for instance, privacy (specifically in cases where parents are divorced), the financing of the treatment (about any financial gain), or responsibility for the whole integrated treatment (who decides when there is a disagreement between professionals).

Some parents would like to have a shared electronic case record for all family members, or if this is impossible because of the rules, they would at least like to give permission for the professionals involved to view both the child’s and the parent’s file. There were also suggestions for a joint front desk where appointments could be better coordinated.

#### 3.4.2. Multi-Disciplinary Consultations

Some parents missed information about the multi-disciplinary consultations. They would like to have known in advance when the multi-disciplinary consultations were scheduled and to receive information afterward about the content of the discussion.

#### 3.4.3. Shared Decision Making

A few parents indicated that they would have preferred to have been more involved in the decision making regarding the options for interventions. For instance, they would have appreciated knowing at the start that there is an individual as well as a group treatment for the parent-child relationship. More importantly, some parents experienced that there was a lack of consensus with their therapist about the focus of treatment. For instance, some parents were struggling with the focus on the parent-child relationship instead of focusing on their child as an individual.

#### 3.4.4. Treatment

Dissatisfaction with some aspects of the treatment included the content and direction of the treatment. This was sometimes related to the insufficiency of addressing their request for help. Some parents expected more assistance at home and were in need of practical advice. In five interviews, the importance of timing was mentioned. For instance, the psychoeducation offered about an autism spectrum disorder was not appropriate to what the parent could manage at that moment. In the parent’s opinion, it should have been scheduled later in the treatment process. Five parents suggested providing a slow phasing out of treatment, by which they meant their need for long-term, low-frequency counseling: 


*“At times, it’s a good thing if you’re still registered [at the mental health service] so you can ask questions, a kind of life coaching. Sometimes a little less, sometimes a little more.”*


#### 3.4.5. Therapeutic Relationship

Some parents felt that they were not being fully understood by their therapist and experienced barriers in the therapeutic collaboration. Specifically, some experienced inequality, poor communication, and a failure to keep appointments.

#### 3.4.6. Parent or Co-Parent

A few parents mentioned some barriers in their own situation in which there was too much stress to start an indicated treatment or to keep their appointments properly. Another barrier that was mentioned was the other parent not being motivated to participate in parent-child therapy.

### 3.5. Modifications to the Theoretical Model of an Integrated Family Approach in Mental Health Care

In our previous study, we presented a theoretical model about key elements of an integrated family approach that professionals consider to be contributors to improved outcomes for the family [13]. A comparison with the key elements mentioned by parents shows that there are three key elements in common: 1. focus on the whole family, 2. flexible and complementary treatment plan tailored for each individual family, and 3. multi-disciplinary consultations. This study revealed three more key elements that parents indicated were contributors to the success of the treatment: 4. components of the whole treatment reinforced each other, 5. the liaison between AMHS and CAMHS, and 6. attention to the social and economic environment. In addition to these key elements, parents mentioned two non-specific elements which, in their view, contributed to treatment success: 1. the therapeutic relationship and 2. the use of videotapes.

When we merge both the professionals’ and parents’ experiences, the resulting theoretical model of the elements contributing to the efficacy of an integrated family approach in mental health care is depicted in Figure 1.

## 4. Discussion

In this study, we evaluated 18 patients and their experiences with an integrated family approach in their treatment at AMHS and with their young child at CAMHS. In all of the included families, there was complexity in the problems they experienced in different domains and treatment that involved a variety of professionals. The aim of this study was to further contribute towards the development of an integrated family approach in adult and child mental health practice for parents and their young children and to contribute to the further development of a theoretical model with respect to the experience of patients as parents.

With respect to our first research question about what patients identify as the improved outcomes of the integrated family approach in their treatment, the analysis of the data revealed that for the majority of the interviewed parents, this approach generates value for themselves and their (relationship with their) young children, as well as for the relationships within the family. A few parents were not able to fully benefit from this integrated treatment; most of them were diagnosed with an autism spectrum disorder. Although this seems notable, the number is too small to draw any conclusions from, other than maybe parents with an autism spectrum disorder need specific attention from professionals offering an integrated family approach in treatment.

One of the benefits of an integrated family approach in treatment, according to patients’ point of view, is that it not only led to improvement in the specific domain toward which the intervention was directed (individual adultchild, parent-child relationship, family relationships and functioning), but, interestingly, it seems that interventions also bring about improvement in other domains. The latter may suggest that domains are closely related to each other, which may create positive cascading results in treatment.

Another interesting result is the similarities and differences in the perceived outcome of an integrated family approach of treatment on the different domains according to parents and professionals. Overall, parents perceived more improvement resulting from the treatment than professionals. They both perceived a similar amount of improvement in the functioning of the adult and the parent-child relationship, but notably, parents experienced substantially more improvement in the family and child domain than the professionals thought had occurred. One possible explanation may be that professionals tend to see the main improvements in those domains where they conducted most of their interventions, while parents do not make these distinctions and have the family as a unit in mind. Another explanation may be that parents with a mental disorder are often worried about what impact their problems and symptoms will have on their children [10], and therefore focused more on improvements in the child’s functioning.

With respect to our second research question about what parents indicate as the key elements of success of an integrated family approach, we found six key elements to an integrated family approach in treatment and two non-specific elements. The key elements were: 1. focus on the family as a whole, 2. flexibility in treatment, tailored to the situation of the family, 3. components of the whole treatment reinforced each other, 4. multi-disciplinary consultation, 5. a liaison between AMHS and CAHMS, and 6. attention to the social and economic environment. The non-specific elements were: 1. the therapeutic relationship and 2. the use of videotapes.

With respect to our third research question, parents brought up some valuable comments about challenges and barriers which posed a threat to the success of an integrated family approach. The most important of these was related to the organization of treatments from two services, with the waiting list being the major barrier. This meant that treatments at AMHS and CAMHS could not be started at the same time, with the result that integrated treatment could not be provided for a period of time. Furthermore, parents noted the importance for them that involved professionals at AMHS and CAHMS be kept well-informed about their situation and the progress of the different treatments. They suggested a joint file or authorization of involved professionals to access the separate files from the two services. Furthermore, they stated the importance of being well-informed regarding what an integrated family approach implies, the scheduling of multi-disciplinary consultations, and the options for interventions. The latter also means that they would have desired a greater voice in the choice of which treatment they preferred. Although not specific to an integrated family approach in treatment, the value of the therapeutic relationship is evident in the criticisms of parents here as well. In their criticism, their wish was for a therapeutic relationship in which they felt equal and were understood, with professionals who transparently communicated, could be trusted to keep appointments, and provided continuity in their relationship. A few patients mentioned barriers related to their own situation or family which limited or prevented benefits from treatment. This is an interesting point because factors in the environment and characteristics of the patients themselves are estimated to contribute 40% to the change process [27,28]. This argues for careful tailoring of the treatment components by professionals to the circumstances and capabilities of the family to prevent them from experiencing disillusionment with treatment, undermining the hope for change.

With respect to our fourth research question, a comparison was made between the key elements of an integrated family approach mentioned by parents and professionals as contributors to treatment success [13]. The results show that there are three key elements in common: focus on the whole family, flexible and complementary treatment plans tailored for each individual family, and multi-disciplinary consultations. This study revealed three more key elements that parents cited as contributors to the success of the treatment. The first, components of the whole treatment reinforced each other, seems to be an important key element because it suggests the following hypotheses: 1. the improved outcome of the whole treatment has more impact than the success of individual treatments separately, and 2. the treatment process with an integrated family approach is more efficient. The second element was that the liaison between AMHS and CAMHS gave them a feeling of familiarity and trust. This seems to be an important element of an integrated family approach in treatment from the patient’s perspective. It suggests that trust in the whole treatment will improve as parents experience connection and consistency among the different treatments provided by AMHS and CAHMS. If this liaison between AMHS and CAMHS is functioning in this way, it may contribute to the establishment of epistemic trust in the whole treatment, which is a key condition for learning and change (see below, [29]). A liaison between AMHS and CAHMS can prevent families from receiving fragmented treatment. Fragmented health systems are considered a major barrier to family-focused practice [30]. Attention to the social and economic environment was the third additional key element mentioned by parents. Although mental health services do not have the expertise and means to intervene in this social and economic context, the number of families in our sample in which there were additional problems in the environment warrants the assessment of this environmental impact given the interaction between the individual, family, and social economic environment domains. The number of families (N = 14) in which other agencies besides the mental health services were involved suggests that inter-agency collaboration and fine-tuning are necessary to prevent overburdening of the family. This is in line with a recently published program theory for family-focused practice [30].

Considering the non-specific elements, many parents described the importance of feeling understood by their therapist and having trust in the therapist. The quality of the therapeutic relationship was mentioned as one of the common factors which contribute to the success of treatment outcomes. In the four-factor model of treatment success, the therapeutic alliance was estimated to account for 30% of the process of change [27,28]. Fonagy and Allison [29] introduced the term “epistemic trust”, which means “an individual willingness to consider new knowledge from another person as trustworthy, generalizable, and relevant to the self” (p. 373). Establishing epistemic trust in the therapeutic relationship between therapist and patient is a prerequisite for the patient to become open to learning, to have new experiences, and to achieve change in perception about oneself, their social relationships, and their behavior. Furthermore, they postulate the process of mentalizing in therapy as a key to epistemic trust, and as a common factor among effective psychotherapies.

Although the therapeutic relationship does not seem to be a specific aspect of an integrated family approach in treatment, it may be meaningful that two-thirds of the parents mentioned this as important for them. Furthermore, dissatisfaction with the therapeutic relationship was mentioned in some cases. Parenthood in general has a potential impact on parents’ feelings of vulnerability, and this is more valid for parents with a mental disorder [10]. It is conceivable that in the treatment of those patients whereby parenthood and the parent-child relationship are included in the treatment, the therapeutic relationship needs specific attention.

The second non-specific element contributing to improved outcomes mentioned by patients was the technique of the use of videotapes to review parent-child interactions. This is in line with previous research [31].

### 4.1. Limitations

Some limitations of this study are similar to the limitations of the previous study which is linked to this one. These concerns regard the sample of families whose treatment was evaluated by the parents. First, the treatment of the families was conducted in a small part of the Netherlands, because there were no other mental health organizations with a comparable liaison between AMHS and CAMHS offering an integrated family approach in treatments to the family. Second, the small sample could be biased in terms of the specific characteristics of the sample.

### 4.2. Implications for Clinical Practice

This evaluation of patients about their experiences with an integrated family approach in the treatment of themselves and with their young child is confirming and complementary to the previous linked study in which the treatment was evaluated by adult and child mental health professionals from their point of view. Together, these studies provide mental health managers and professionals guidance regarding which key elements and processes are essential if adult and child mental health professionals are to offer an integrated family approach in their treatments for the benefit of their patients as a parent and their families. More importantly, an integrated family approach in mental health care supports parents in parenthood and could prevent both parents and their children from the consequences of the risks related to the parental mental disorder.

## 5. Conclusions

The majority of patients as parents in this study indicate that they experienced the integrated family approach in their treatment to be of benefit to themselves, their young children, and the relationships in their family, especially the parent-child relationship. Improved outcomes were described in the domains of the family, the parent-child relationship, individual symptoms, and the functioning of the parent and the child as individuals. Patients mentioned six key elements which contribute to the improved outcome of the integrated family approach in treatment. The first three correspond to what professionals have mentioned as the key elements of success: focus on the whole family, flexibility and treatment tailored to the situation of the family, and multi-disciplinary consultation between professionals. The additional key elements according to patients were that components of the whole treatment reinforced each other, a liaison between adult and child mental health services, and attention to the social and economic environment. The latter implies that close collaboration with social services is required. Furthermore, the quality of the therapeutic relationship and the use of videotapes in treatment seem to be important factors in success. Both the key elements and the recommendations related to the barriers experienced by patients in their treatments may be conceived as contributing to improving the treatment for families in the practice of adult and child mental health care.

## Figures and Tables

**Figure 1 ijerph-19-13164-f001:**
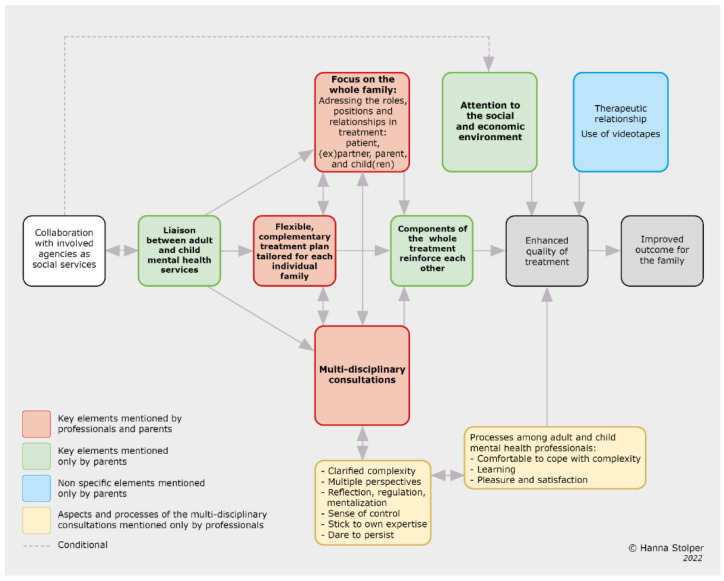
Model of an integrated family approach in mental health care according to professionals and parents.

**Table 1 ijerph-19-13164-t001:** Characteristics of adult patients (N = 18) and their children (N = 18).

		N	%
Gender adult	Man	3	16.7
	Woman	15	83.3
Gender child	Boy	10	55.6
	Girl	8	44.4
Age (adult patients, years)	<30	4	22.2
	30–35	8	44.4
	>35	6	33.3
Age (child, months)	0–12	9	50.0
	13–36	5	27.8
	37–72	4	22.2
Highest educational level attained (adult patient)	Low (basic or pre-vocational secondary education)	3	16.7
	Middle (secondary vocational education)	11	61.1
	High (bachelor’s or master’s degree)	4	22.2
Raised as a child by (adult patients)	Biological parent (s)	11	61.1
	Adoptive parents	2	11.1
	Foster parents or institution	5	27.8
Grew up with (child)	Both biological parents	12	66.7
	One biological parent	4	22.2
	Post-divorce co-parenting	1	5.6
	Foster parents	1	5.6
Classification DSM-5 (adult patients)	Personality Disorder	5	27.8
	Bipolar Disorder	1	5.6
	Depressive Disorder	3	16.7
	Anxiety Disorder	2	11.1
	Autism Spectrum Disorder	3	16.7
	Post-Traumatic Stress Disorder	3	16.7
	Other Specified Trauma and Stressor-Related Disorder	1	5.6
	Comorbidity	12	66.7
Classification DSM-5 (child)	Autism Spectrum Disorder	1	5.6
	Unspecified Neurodevelopmental Disorder	3	16.7
	Post-Traumatic Stress Disorder	2	11.1
	Parent-Child Relational Problem	12	66.7
	Comorbidity	7	38.9
Number of previous treatments in mental health care (adult patients)	First treatment	4	22.2
	Second treatment	2	11.1
	More than two treatments before	12	66.7
Duration of treatment with an integrated family approach (months)	0–6	0	0.0
	6–12	2	11.1
	12–24	9	50.0
	>24	7	38.9

**Table 2 ijerph-19-13164-t002:** Treatment interventions in different domains.

Domain	Treatment Interventions	N = 18	
		N	%
Individual patient: adult	Psychiatric nursing	9	50
	Psychotherapy	9	50
	Systems Training for Emotional Predictability and Problem Solving (STEPPS)	1	5.6
	Pharmacotherapy	11	61.1
	Eye Movement Desensitization and Reprocessing (EMDR)/Imaginal exposure	4	22.2
	Psychomotor therapy	1	5.6
Parent-child relationship	Parent-child psychotherapy	13	72.2
	Parent counseling	9	50
	Parent-child group	4	22.2
Family	Couples therapy	6	33.3
	Home treatment	4	22.2
	Family therapy	4	22.2
Individual patient: child	Eye Movement Desensitization and Reprocessing/Storytelling	2	11.1
	Pharmacotherapy	1	5.6
	Psychoeducation autism spectrum disorder	1	5.6

**Table 3 ijerph-19-13164-t003:** Improved outcomes in different domains perceived by parents.

Domain	Improved Outcome by Domain	
		N
Individual patient: adult	Decrease in symptoms of the mental disorder Regulation of emotions and behavior	15
Parent-child relationship	Quality of the relationship Empathize and attune with the child	15
Family	(Ex)couple-relationship Parental collaboration Parenting skills Confidence in parenthood Family relations Extended family relationships	16
Individual patient: child	Decrease in symptoms of the mental disorder Relationships Emotion regulation	10

**Table 4 ijerph-19-13164-t004:** Improved outcome of multidisciplinary treatments of an integrated family approach in mental health care according to parents and professionals.

Improved Outcome of Treatment			According to Patients/Parents		According to Professionals	
Treatments Performed (N = 18)						
Domains	N	%	N	%	N	%
Individual patient	17	94.4	15	83.3	15	83.3
Parent-child relationship	15	83.3	15	83.3	15	83.3
Family domain	13	72.2	16	88.9	9	50
Child domain	4	22.2	10	55.5	3	16.7

**Table 5 ijerph-19-13164-t005:** Key elements and nonspecific elements which contributed to the success of the treatment according to parents.

Key Elements of an Integrated Family Approach in Treatment
Focus on the family as a whole
Flexibility in treatment, tailored to the situation of the family
Components of the whole treatment reinforced each other
Multi-disciplinary consultations
A liaison between AMHS and CAHMS
Attention to the social and economic environment
**Nonspecific elements**
Therapeutic relationship
Use of videotapes

## Data Availability

The raw data will be available on request.

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
