# Peer review of "The Patient’s Voice as a Parent in Mental Health Care: A Qualitative Study"

_ijerph, 2022, doi:10.3390/ijerph192013164_

Round 1

Reviewer 1 Report

This research presents the perspective of parents with mental health issues who are being cared for in an integrative family program along with their children. This article is the continuation of a larger study to build a theoretical framework about integrated family care. In the previous study, the perspective of professionals was analyzed.

The introduction and background seem complete and clear to me. 

I have some doubts about the methodology used. Within the framework of qualitative research, it is said that the case method with thematic analysis has been used. On the other hand, the objective is to deepen the integrated family care program from the perspective of patients. This could also fit with a phenomenology, or a grounded theory, since data is used to general a theoretical framework. I have doubts about whether there has been an in-depth reflection on the specific qualitative method to be used in the research. In any case, being a qualitative approach, there are several elements of the results, discussion and limitations section that I suggest modifying.In any case, being a qualitative approach, there are several elements of the results, discussion and limitations section that I suggest modifying.

In my opinion, section 3.1 of the results is quantitative, not qualitative. I believe that the explanation of the treatment interventions should be explained in the methodology, eliminating Table 2, and that this section should be the description of the outcomes perceived by the patients,  incorporating verbatim. 

Likewise, in the entire results section I suggest eliminating references to the percentage of patients who speak or comment on each topic. It is NOT a random sample that allows the generalization of results, it is a sample for a qualitative study that allows the extrapolation of the results to similar contexts. Therefore, having reached data saturation, in my opinion it does not matter whether 50% or 70% have said so.

The numbering of the subsections of the results must be reviewed. They are wrong. The 3.3. it's really 3.2.1 and so on.

Despite these problems with the presentation of the results, the results obtained are clear and Figure 1 summarizes the theoretical model very clearly.

The discussion seems complete to me. But in the limitations section I suggest removing the part about the random sample. The characteristics of qualitative sampling are not a limitation. 

In short, I think it is good research, but I get the feeling that researchers are not completely familiar with qualitative research and unconsciously try to transfer aspects of quantitative research to the manuscript (percentages, consider qualitative sampling a limitation, etc ...). I think that with some changes (simple but relevant) the quality can be improved.

Author Response

Dear Reviewer,

See attachment. 

With kind regards, 

Hanna Stolper

Reviewer 2 Report

The need for the study is sound and well justified. The research questions are comprehensive and sound. The purpose and methods are well described and justified. The findings (note qualitative research has Findings not Results) are generally sound; however, more descriptions are needed regarding the various treatments offered to parent/s, child, and  family. In addition, more discussion is needed regarding the different reports from parents and professionals regarding the types of improvements noted in child and also family (Line 287-295).  

The implications for clinical practice (p. 18) are sound but could add more discussion - for example, regarding the changes to try.

Smaller Suggestions: Sentence structure needs changes on lines: 46, 87, 556, 639, 727, 738. Also need to spell out EMDR (line 291) - i.e., Eye Movement Desensitization and Reprocessing.    

Author Response

(The authors gave the same response as above.)

Round 2

Reviewer 1 Report

The changes made have improved the quality of the manuscript. Congratulations!